# Structural modelling and preventive strategy targeting of WSSV hub proteins to combat viral infection in shrimp *Penaeus monodon*

Tanate Panrat[1,2], Amornrat Phongdara[2], Kitti Wuthisathid[3], Watcharachai Meemetta[3], Kornsunee Phiwsaiya[3,4], Rapeepun Vanichviriyakit[3,5], Saengchan Senapin[3,4], Pakkakul Sangsuriya[6]*

**1** Prince of Songkla University International College, Prince of Songkla University, Hatyai Campus, Songkhla, Thailand, **2** Center for Genomics and Bioinformatics Research, Faculty of Science, Prince of Songkla University, Songkhla, Thailand, **3** Center of Excellence for Shrimp Molecular Biology and Biotechnology (Centex Shrimp), Faculty of Science, Mahidol University, Bangkok, Thailand, **4** National Center for Genetic Engineering and Biotechnology (BIOTEC), National Science and Technology Development Agency (NSTDA), Pathum Thani, Thailand, **5** Department of Anatomy, Faculty of Science, Mahidol University, Bangkok, Thailand, **6** Aquatic Molecular Genetics and Biotechnology Research Team, BIOTEC, NSTDA, Pathum Thani, Thailand

* pakkakul.san@biotec.or.th

**Data Availability Statement:** All relevant data are within the manuscript and its Supporting Information files.

## Abstract

White spot syndrome virus (WSSV) presents a considerable peril to the aquaculture sector, leading to notable financial consequences on a global scale. Previous studies have identified hub proteins, including WSSV051 and WSSV517, as essential binding elements in the protein interaction network of WSSV. This work further investigates the functional structures and potential applications of WSSV hub complexes in managing WSSV infection. Using computational methodologies, we have successfully generated comprehensive three-dimensional (3D) representations of hub proteins along with their three mutual binding counterparts, elucidating crucial interaction locations. The results of our study indicate that the WSSV051 hub protein demonstrates higher binding energy than WSSV517. Moreover, a unique motif, denoted as "S-S-x(5)-S-x(2)-P," was discovered among the binding proteins. This pattern perhaps contributes to the detection of partners by the hub proteins of WSSV. An antiviral strategy targeting WSSV hub proteins was demonstrated through the oral administration of dual hub double-stranded RNAs to the black tiger shrimp, *Penaeus monodon*, followed by a challenge assay. The findings demonstrate a decrease in shrimp mortality and a cessation of WSSV multiplication. In conclusion, our research unveils the structural features and dynamic interactions of hub complexes, shedding light on their significance in the WSSV protein network. This highlights the potential of hub protein-based interventions to mitigate the impact of WSSV infection in aquaculture.

**Funding:** Mahidol University (Fundamental Fund: fiscal year 2023 by the National Science Research and Innovation Fund (NSRF), Grant no. FF-056/2566 and the NSRF and Prince of Songkla University (Grant no. UIC6601175S).

**Competing interests:** The authors have declared that no competing interests exist.

## Introduction

White spot disease (WSD) is an economically damaging viral disease that deteriorates the shrimp culture industry throughout the world. Since WSD outbreaks result in a high mortality rate of shrimp of up to 100% within 3-10 days [1], they have caused losses in shrimp production. The causative agent of the disease is white spot syndrome virus (WSSV), which is a member of the genus *Whispovirus*, family *Nimaviridae*. WSSV is a rod-shaped to elliptical, circular double-stranded DNA (dsDNA) virus. Its 300-305 kbp genome was first completely characterized in different isolates from Thailand, China, and Taiwan, and then many more WSSV isolates, such as Korea, Mexico, Ecuador, and Australia, were reported [2]. Numerous studies have aimed to focus on unraveling the complexities of WSSV protein interactions, as this endeavor is crucial for advancing the understanding of viral infections, developing effective therapeutic strategies, and enhancing the ability to control and prevent viral diseases.

Several investigations on protein-protein interactions in WSSV have provided insights into viral morphogenesis by examining the interactions among structural proteins of the virus. For example, the major envelope protein VP24 was discovered to interact with VP19, VP26, and VP28 [3, 4]. VP26 was also recognized as a linker protein in the assembly of a VP51A-VP26-VP28 complex [5]. VP11 was found to have direct interactions with 12 structural proteins of WSSV as well as with itself [6]. Furthermore, a comprehensive network of protein-protein interactions (PPIs) within WSSV was mapped at a proteomic scale [7]. This was achieved through a yeast two-hybrid approach involving around 200 WSSV proteins. The study identified the top eight proteins (WSSV004, WSSV051, WSSV118, WSSV188, WSSV349, WSSV395, WSSV471, and WSSV517) as hub proteins, each demonstrating a substantial number of interaction partners ranging from 20 to 48 [7]. Two hub proteins, WSSV051 and WSSV517, have been confirmed to interact with various binding partners using co-immunoprecipitation. WSSV051 was previously annotated as a structural protein VP55, as indicated by proteomic analysis of purified WSSV virions [8], while WSSV517 was identified as an early protein gene through DNA microarray analysis [9]. In the WSSV PPI network, mutual binding partners of these two hubs include WSSV144, WSSV322, and WSSV454. The function of WSSV144 is unknown, while WSSV322 is an anti-apoptotic protein and WSSV454 as a thymidine-thymidylate kinase protein (TK-TMK). An *in vitro* insect model has demonstrated that WSSV322 binds with the shrimp caspase protein and exhibits anti-apoptotic activity [10]. The homology similarity and thymidine kinase activity confirm WSSV454's function as a TK-TMK protein [11, 12]. This suggests that two hub proteins are likely involved in several biological processes related to the functions of their binding partners, including viral assembly, DNA replication, and nucleotide metabolism [7]. Although the precise interaction mechanisms of these hubs in viral biology remain uncertain, it has been demonstrated that they play crucial roles in the WSSV life cycle. This is evident from the fact that suppressing WSSV051 or WSSV517 resulted in a delay in shrimp mortality caused by WSSV infection [7, 13]. However, more investigation into the complexity of WSSV hubs and their mutual interacting partners is still required to comprehend their properties, which could lead to their use as antiviral targets.

Understanding the structural characteristics of WSSV hubs and their interacting partners is currently limited due to challenges or infeasibility in obtaining three-dimensional (3D) structures through experimental techniques such as X-ray crystallography and nuclear magnetic resonance (NMR). Protein computer simulations offer a valuable alternative for exploring the properties and relationships of essential amino acids within protein sequences, as well as their 3D structures and functions in biological systems. Thus, this study focuses on presenting evidence through docking simulations of interactions among WSSV proteins. Two hubs,

WSSV051 and WSSV517, were selected from the PPI network based on previous data identifying them as targets for gene silencing to control WSSV infection in shrimp [7, 13]. Specifically, WSSV144, WSSV322, and WSSV454, identified as mutual interacting partners of both WSSV051 and WSSV517, were chosen for analysis. For the first time, the study investigated the 3D structures and binding characteristics of these hubs with their three mutual partners. Furthermore, the current research evaluated a potential strategy for preventing WSSV infection in shrimp by concurrently knocking down these two hubs through the oral administration of combined dsRNA targeting these specific hubs.

## Materials and methods

### Sequence analysis and functional domain prediction of WSSV proteins

In this study, two selected WSSV hub proteins (WSSV051 and WSSV517) and their three mutual binding partners (WSSV144, WSSV322, and WSSV454) were elucidated through *in silico* analysis. WSSV sequences (S1 Table in S2 File) retrieved from the NCBI database were analyzed for functional domains, disordered and pore lining regions and phosphorylation sites by SMART [14], PSIPRED [15], the NetPhos 3.1 server [16] and the ViralPhos server [17], respectively.

### *In silico* simulation of WSSV hub proteins and their interacting proteins

Since most WSSV proteins lack X-ray or NMR solution structures, as in the RCSB Protein Data Bank (www.rcsb.org), the three-dimensional (3D) coordinate templates of WSSV in the present study can be obtained by integrating various simulation methods, including threading based on LOMETS (https://zhanglab.ccmb.med.umich.edu/LOMETS) [18], pDomTHREADER searching on the PSIPRED server (http://bioinf.cs.ucl.ac.uk/psipred) [15, 19, 20], and homology modeling on the SWISS-MODEL server (https://swissmodel.ExPASy.org) [21–23]. A list of selected templates is shown in S3 Table in S2 File. Subsequently, the 3D structures of WSSV hubs and their interacting proteins were determined with the I-TASSER server [24]. The quality of the predicted structures was validated by the ProFunc server [25]. The molecular docking simulation for estimating the feasible interaction sites of WSSV complexes was then performed using the ClusPro 2.0 server (https://cluspro.bu.edu) [26], AutoDock tool (http://autodock.scripps.edu) [27, 28] and PyMol [29, 30]. Two approaches, including "one-on-one docking simulation", where a single molecule of receptor docks with a single ligand, and "competitive docking simulation", where a single receptor docks with several ligands, were performed in this study. For one-on-one docking simulation, the predicted 3D models of WSSV051 and WSSV517 were assigned as receptor molecules and the binding proteins (WSSV144, WSSV322, and WSSV454) as ligand proteins. The first ranking of the simulated complex with the best center and lowest binding energy score was selected to determine the binding sides in detail. The competitive docking simulation was also examined by setting the hub proteins as receptors and interacting proteins as ligands. In round one of the competitive docking map, the 1st ligand was docked with the hub receptor. The best docking complex of round one was subsequently applied as a receptor in round two, where the 2nd ligand was then added. In round three, the last fragment or 3rd ligand was also employed as the ligand to bind with the previous complex.

### Production of double-stranded (ds) RNA in the bacterial system

In a previous study, a hairpin-dsRNA expression plasmid targeting one of the hubs, WSSV051, was already constructed [13]. The present study aimed to investigate the effect of combined

**Table 1. Primers used in this study.**

| Primer name | Sequence (5' to 3') | Usage | Reference |
|---|---|---|---|
| WSSV051-F | TTCACGAACGGCTGCCATTT | Transcription analysis | [7] |
| WSSV051-RNAi-R | GCGGTAGCGTTCTCTTCATC | | |
| WSSV517-RNAi-F | AAGGAACGGAATGTCCACAA | dsRNA production & Transcription analysis | [7] |
| WSSV517-RNAi-R2 | TTACAACATAATTACTTGCC | | This study |
| *Xba*I-WSSV517-RNAi-F | GCTCTAGAAAGGAACGGAATGTCCAC | | |
| *Hin*dIII-WSSV517-RNAi-R3 | ACAAGCTTCCAGTGGTGCTGACGATG | | |
| WSSV517-R | TCCCCGCGGTTTTGTTCCTTGTAATT | | [7] |
| VP28-F | AGGTGTGGAACAACACATCAAG | WSSV detection | [58] |
| VP28-R | TGCCAACTTCATCCTCATCA | | |
| Actin-F | AGGCTCCCCTCAACCCCAAGG | Transcription analysis | [7] |
| Actin-R | GCAGTGATTCTGCATGCG | | |

hub dsRNA, targeting WSSV051 and WSSV517. Thus, the previously made WSSV051 construct was used, and a new hairpin-dsRNA expression plasmid targeting WSSV517 was created in this study. The RNAi capacity region in WSSV517 was predicted accordingly [31], and its sense orientation was first amplified from WSSV-infected shrimp DNA using the specific primers WSSV517-RNAi-F and WSSV517-RNAi-R2 (Table 1) and then cloned under the T7 promoter of the pDrive vector (QIAGEN). The antisense strand was obtained using primers harboring *Xba*I and *Hin*dIII restriction sites, *Xba*I-WSSV517-RNAi-F and *Hin*dIII-WSSV517-RNAi-R3 (Table 1), and then ligated into a linearized recombinant plasmid of the sense strand. The hairpin construct of WSSV517 was verified by DNA sequencing. Each recombinant plasmid was transformed into RNase III-deficient *Escherichia coli* HT115 DE3. To produce dsRNA, the overnight culture of a single transformant colony in LB medium containing 100 μg/ml ampicillin and 12.5 μg/ml tetracycline, cultured at 37°C with an $OD_{600}$ of 0.4, was induced with 0.5 mM IPTG. The culture was grown for 4 h and then collected for further dsRNA extraction using the phenol–chloroform method. To remove single-stranded (ss) RNA and DNA, the nucleic acid contents were subsequently treated with RNase A and DNase I, respectively, followed by dsRNA purification using the phenol–chloroform method. The amount of dsRNA was assessed by measuring absorbance at 260 and 280 nm. The characteristics of dsRNA were verified by nuclease assays in which it was treated with RNase III, RNase A and DNase I, which specifically digested dsRNA, ssRNA and DNA, respectively. All treated reactions were analyzed by gel electrophoresis along with untreated dsRNA as a control.

## Preparation of shrimp feed containing WSSV hub dsRNA

Commercial shrimp feed (36% protein; CP) was prepared to contain either 60 mg or 120 mg of WSSV hub dsRNA per 1 kg of feed. The two formulations were (I) 30 mg each of WSSV051 and WSSV517 dsRNA and (II) 60 mg each of WSSV051 and WSSV517 dsRNA. These dsRNA dosages were 10 times higher than those used in our previous study [13], which utilized 6 mg and 12 mg, with the aim of enhancing protection against WSSV infection. Briefly, ground commercial feed was thoroughly mixed with bacterial suspension, passed through a syringe and then dried at 60°C overnight. The dried feed was then cut into small pellets and kept at room temperature. Bacterial cells with no expression plasmid were also cultured and mixed with shrimp feed to serve as a control feed formulation. To investigate dsRNA stability after feed preparation, feed pellets (~ 0.1 g) at 3 – 10 weeks post feed preparation were subjected to RNA extraction using TRIzol™ (Invitrogen) according to the manufacturer's instructions.

The presence of dsRNA was subsequently examined by RT–PCR using the primer pairs WSSV051-F and WSSV051-RNAi-R for WSSV051 dsRNA or WSSV517-RNAi-F and WSSV517-R for WSSV517 dsRNA (Table 1). RT–PCR was carried out in a 20 µl reaction solution containing dsRNA template, 0.2 µM of each forward and reverse primer, 0.8 µl of Super-Script One-Step RT/Platinum Taq mix (Invitrogen), and 1X reaction buffer. Reverse transcription was performed at 50˚C for 30 min and 94˚C for 2 min, followed by 35 cycles of 94˚C for 15 s, 55˚C for 30 s, and 68˚C for 30 s. RT–PCR products were analyzed by agarose gel electrophoresis.

## Oral delivery of WSSV hub dsRNA and WSSV challenge

Shrimp *Penaeus monodon* (10 g body weight) were obtained from a local farm and were acclimatized in a wet laboratory in 1000 liters aquaria containing continuously aerated artificial seawater at 20 ppt and 28˚C for 7 days before the experiments began. Meanwhile, gill tissues from cultured shrimp were randomly subjected to WSSV PCR detection to ensure that shrimp were WSSV-free. The 120 shrimp were divided into four groups (n=30) in which two tested groups, I and II, were fed 60 mg combined dsRNA feed and 120 mg combined dsRNA feed, respectively. Meanwhile, two control groups, III and IV, were fed feed containing non-expressed dsRNA bacterial cells. All shrimp groups were fed twice a day with a feeding dose of 3% of shrimp body weight. After feeding for 3 days, groups I, II and III were injected intramuscularly with 50 µl of WSSV stock using a sterile 1-ml syringe with a 26-gauge needle. This resulted in approximately 50% shrimp death within 5 days (5-day $LD_{50}$). This WSSV inoculum was prepared as described previously [7]. Shrimp in group IV was injected intramuscularly with 150 mM NaCl as a negative control. All shrimp groups were continuously fed with the feed formulas as described above throughout the entire 7-day experiment. Cumulative mortality in each group was monitored twice a day. Shrimp specimens in each group (n=3) were also sampled at 6 days post WSSV injection for determination of gene knockdown efficiency and WSSV infection (see below). All shrimp used in this experiment were approved by the National Center for Genetic Engineering and Biotechnology IACUC (Project Code BT-Animal 25/2567).

## Gene silencing efficiency of combined WSSV hub dsRNA

Total RNA of shrimp specimens was isolated from gill tissues using TRIzol™ (Invitrogen) as described in the manufacturer's protocol. Semi-quantitative RT–PCR was performed to detect WSSV051 and WSSV517 transcripts using the respective primer pairs WSSV051-F and WSSV051-RNAi-R and WSSV517-RNAi-F and WSSV517-R (Table 1). The partial *β-actin* gene was also amplified as an internal control. RT-PCRs were carried out in a 20 µl reaction solution containing total RNA template, 0.2 µM of each forward and reverse primer, 0.8 µl of SuperScript One-Step RT/Platinum Taq mix (Invitrogen) and 1X reaction buffer. The reaction protocol comprised reverse transcription at 50˚C for 30 min followed by denaturation at 94˚C for 2 min followed by PCR cycling consisting of denaturation at 94˚C for 15 s, annealing at 55˚C for 30 s and extension at 68˚C for 30 s. The PCR cycling was optimized to detect each transcript as follows: 35 cycles for WSSV genes and 25 cycles for β-actin. RT–PCR products were analyzed by agarose gel electrophoresis.

## WSSV diagnosis by PCR

WSSV infection was examined by PCR amplification of VP28 using the specific primer pairs listed in Table 1. DNA samples were prepared from gill tissues using TF lysis buffer as described previously [7]. PCRs were carried out in a 20 µl reaction solution containing DNA

sample, 0.2 μM of each forward and reverse primer, 200 μM of dNTP mix, 1.5 mM of MgCl₂, 2.5 U of *Taq* DNA polymerase (Invitrogen), and 1X reaction buffer. The amplification protocol comprised first denaturation at 94°C for 3 min followed by 30 cycles of denaturation at 94°C for 30 s, annealing at 55°C for 30 s and extension at 72°C for 30 s. Amplification of shrimp β-actin was carried out using the same reaction components described for VP28 above with a 25 thermal cycling protocol instead of serving as an internal control.

## Results

### Sequence analysis and functional domain prediction of WSSV proteins

The SMART analysis results showed that WSSV051 contains low-complexity regions (LCRs) at amino acid positions 92–103, 280–292, and 312–326, whereas WSSV517 has a transmembrane signal at amino acid positions 2–24 of the N-terminal region. WSSV454 shows a high confidence feature of thymidine kinase at the N-terminus (position 3–182), a low complexity region (position 183-193), and thymidylate kinase at the C-terminus (position 197–378). WSSV322 contains the low complexity region at positions 5–25, 35–54, and 221–232, whereas WSSV144 does not show any significant features (S1 Table in S2 File). PSIPRED-MEMSAT secondary structure prediction revealed pore-lining areas in WSSV051 (at position 232–247), WSSV454 (at position 279–294), WSSV322 (at position 60–75), and WSSV144 (at position 16–31), while WSSV517 lacked such regions. Moreover, PSIPRED-MEMSAT also displays the disordered protein binding region in WSSV051 at positions 1–10 and 283–285, WSSV517 at 1–13, WSSV322 at 1–17, 35–49, and 275–289, WSSV144 at 1–8 and 53–75, and no disordered protein binding region in WSSV454 (S1 Table in S2 File). Furthermore, most of the phosphorylation sites of WSSV proteins were predicted at serine residues, followed by threonine and tyrosine residues. Nevertheless, tyrosine phosphorylation sites were not observed in WSSV144 and WSSV454. The summary result of predicted phosphorylation sites is illustrated in S2 Table in S2 File.

Apart from protein sequence analysis, distinctive protein motifs of "S-G-x(2)-S-x(2)-T-x(2)-N-S" and "N-S-x(1,2)-V-G-x-L-x(5)-D" can be found in both WSSV051 and WSSV517; however, their roles are still unknown. The motif "S-G-x(2)-S-x(2)-T-x(2)-N-S" is revealed in WSSV051 at Ser157–Ser168 and WSSV517 at Ser32–Ser43, whereas the motif "N-S-x(1,2)-V-G-x-L-x(5)-D" is located in WSSV051 at Asn172–Asp184 and WSSV517 at Asn46–Asp58. Interestingly, the hub binding proteins WSSV144 (at Ser57–Pro67), WSSV322 (at Ser225–Pro235), and WSSV454 (at Ser67–Pro77) were found to have a highly similar sequence pattern with the motif "S-S-x(5)-S-x(2)-P". Additionally, certain repeated amino acid sequences were also discovered in WSSV proteins. The repeating regions of "F-x(3)-E-x(2)-S-x(3)-I" were found at Phe116–Ile177 and Phe194–Ile205 in WSSV051, and the repeating regions of "P-x(3)-I-x(4)-D-x(3)-P-x(2)-P-x-L" were located at Pro79–Leu66 and Pro93–Leu107 in WSSV322. In WSSV454, the repetitive "V-x-H-x(2)-E-x(3)-G-C-x-T" regions were found at Val37–Thr49 and Val53–Thr65.

### Structural analysis of WSSV proteins

The predicted 3D structures of WSSV051 and WSSV517 hubs were revealed to form a beta-propeller blade shape, as shown in Fig 1A and 1B, respectively. The predicted structure model of WSSV051 contains thirteen β-sheets and three α-helices, whereas WSSV517 shows eighteen β-sheets and one α-helix near the central axis of the predicted protein structure. The quality of the predicted structures was validated by Ramachandran analysis on the ProFunc server. The Ramachandran plot of WSSV051 indicated that 58.97% of amino acid residues were present in the most favored regions, 29.74% of residues in additional allowed regions, 7.70% of residues

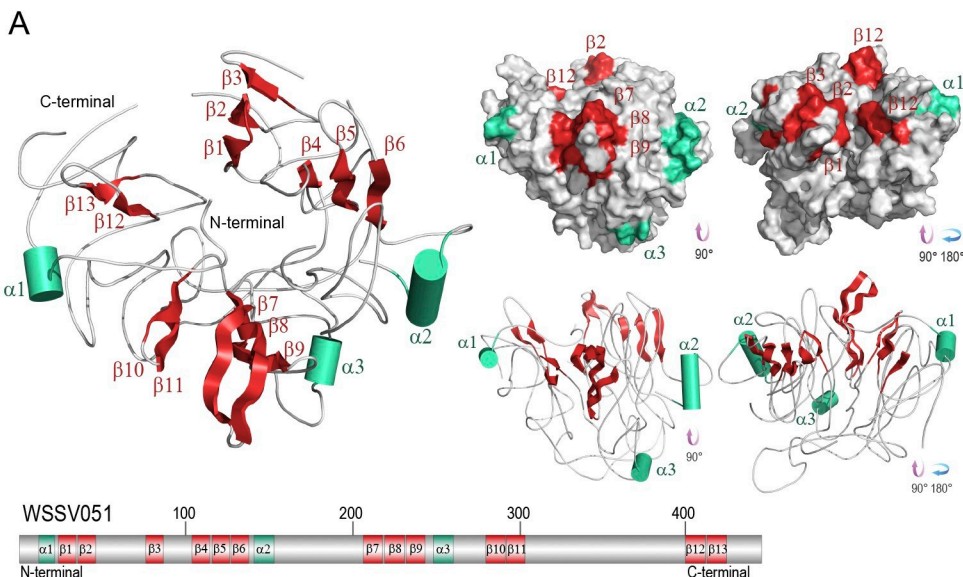

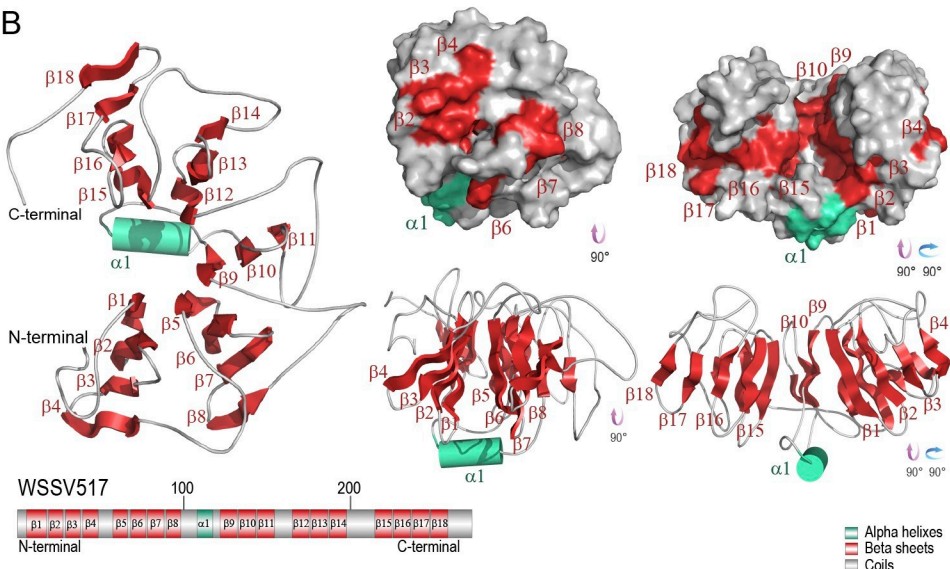

**Fig 1. Structural analysis of WSSV hub proteins.** The predicted 3D models of WSSV051 (A) and WSSV517 (B) were generated by using homology modeling on the SWISS-MODEL server combined with threading-based LOMETS on the I-TASSER server and the pDomTHREADER method on the PSIPRED server. The structural element of α-helices is shown in turquoise, β-sheets in maroon, and coils in pearl. Each of the β-sheets of WSSV051 and WSSV517 are twisted to form a β-propeller blade shape.

in generously allowed regions, and 3.59% of residues in disallowed regions (S1 Fig in S1 File). For WSSV517, 53.19% of residues were found in the most favored regions, 40.00% of residues in additional allowed regions, 3.40% of residues in generously allowed regions and 3.40% of residues in disallowed regions (S1 Fig in S1 File). Likewise, 3D models of WSSV144, WSSV322, and WSSV454 were built, and the quality structures were validated, as shown in S2–S4 Figs in S1 File. The predicted 3D model of WSSV144 showed only a coiled structure, whereas WSSV322 represented ten α-helices. Moreover, the structural analysis of WSSV454

revealed seven β-sheets and twelve α-helices, which found a disulfide bridge between Cys141 and Cys178 and folded into a horseshoe shape.

## Docking simulation of WSSV hubs and interacting proteins

For the one-on-one docking simulation of WSSV051 and the binding partners (WSSV051: WSSV144, WSSV051:WSSV322, and WSSV051:WSSV454), WSSV051:WSSV454 showed a higher binding energy score of interaction than WSSV051:WSSV322 and WSSV051: WSSV144. The center binding energy and lowest binding energy of WSSV051:WSSV454 were -1144.30 Kcal/mol and -1157.00 Kcal/mol, respectively, whereas those of WSSV051:WSSV322 were -1081.10 Kcal/mol and -1204.30 Kcal/mol, and those of WSSV051:WSSV144 were -850.70 Kcal/mol and -1032.20 Kcal/mol (Table 2). Fig 2 shows a graphic illustration of the WSSV051 structure interacting with partners. The analysis of WSSV051 docking with WSSV454 (Fig 2A) revealed that the amino acid residues Glu46–Ile85 and Ile111–Leu141 of WSSV051 are bound to amino acid residues Gly10–Leu20, Leu35–Glu61, and Lys136–Val195 of WSSV454. For WSSV051:WSSV322 (Fig 2B), WSSV051 interacts with both the N-terminal (Met1–Glu50 and Ser95–Pro105) and C-terminal (Ser195–Leu281) regions of WSSV322. It was found that Met311–Glu348 in the low-complexity region and Glu29–Asp80 of WSSV051 contributed to the interaction with WSSV322. The binding region analysis of the WSSV051: WSSV144 complex (Fig 2C) also showed that the four binding sites of WSSV051, Gly216–His257, Met283–Leu296, Arg361–Pro370, and Ile386–Leu393, interacted with WSSV144 at N-terminal Met1–Val15 and C-terminal Gln66–Tyr80. The important binding domains that facilitate WSSV454, WSSV322, and WSSV144 attachment to the WSSV517 hub protein were also discovered (Fig 3). The simulated complex of WSSV517:WSSV454 (Fig 3A) indicated that three amino acid regions at Ile10–Gly48, Asn121–His133 and Lys221–Gly263 of WSSV517 interact with WSSV454 at Gly15–Ile85, Thr131–Ile146, and Ile166–Cys178, with a center binding energy of -900.00 Kcal/mol and the lowest binding energy of -973.70 Kcal/mol (Table 2). For the WSSV517:WSSV322 complex (Fig 3B), the N-terminus of WSSV517 at Met1–Ala12, Ile41–Ser61, and Thr86–Ser95 was demonstrated to interact with WSSV322 at Gly151–Tyr197. Indeed, the Thr86–Ser95 region in WSSV517 is likely to accommodate Gly151–Tyr197 and Ser226–Glu240 in WSSV322. The amino acid sequences Gly15–Ile85 of WSSV454 and Ser226–Glu240 of WSSV322, which were used for binding with WSSV517, like- wise include the motif "S-S-x(5)-S-x(2)-P". This suggests that the motif "S-S-x(5)-S-x(2)-P" may have a role in hub recognition. For the WSSV517:WSSV144 simulation complex (Fig 3C), it was suggested that WSSV144 with amino acid residues Met1–Gly31, Thr51–Ser56, and Arg71–Tyr80 may facilitate WSSV144 binding with WSSV517 at Met1–Gly50, Arg120– Lys131, and Lys221–Lys265. The center binding energy and the lowest binding energy of WSSV517:WSSV144 are -940.30 kcal/mol and -1157.40 kcal/mol, respectively, whereas the

**Table 2. The putative binding sites of WSSV hub proteins and their binding partners.**

| Predicted possible binding sites | | | | Binding energy scores (Kcal/mol) | |
|---|---|---|---|---|---|
| Hub Proteins (Accession number) | Amino acid residues | Binding Partners (Accession number) | Amino acid residues | Center | Lowest |
| WSSV051 (AAL88919.1) | 46–85, 111–141 | WSSV454 (AAL89322.1) | 10–20, 35–61, 136–195 | -1144.30 | -1157.00 |
| | 29–80, 311–348 | WSSV322 (AAL89190.1) | 1–50, 95–105, 195–281 | -1081.10 | -1204.30 |
| | 216–257, 283–296, 361–370, 386–393 | WSSV144 (AAL89012.1) | 1–15, 66–80 | -850.70 | -1030.20 |
| WSSV517 (AAL89385.1) | 10–48, 121–133, 221–263 | WSSV454 (AAL89322.1) | 15–85, 131–146, 166–178 | -900.00 | -973.70 |
| | 1–12, 41–61, 86–95 | WSSV322 (AAL89190.1) | 151–197, 226–240 | -783.20 | -799.30 |
| | 1–50, 120–131, 221–265 | WSSV144 (AAL89012.1) | 1–31, 51–56, 71–80 | -840.30 | -1157.40 |

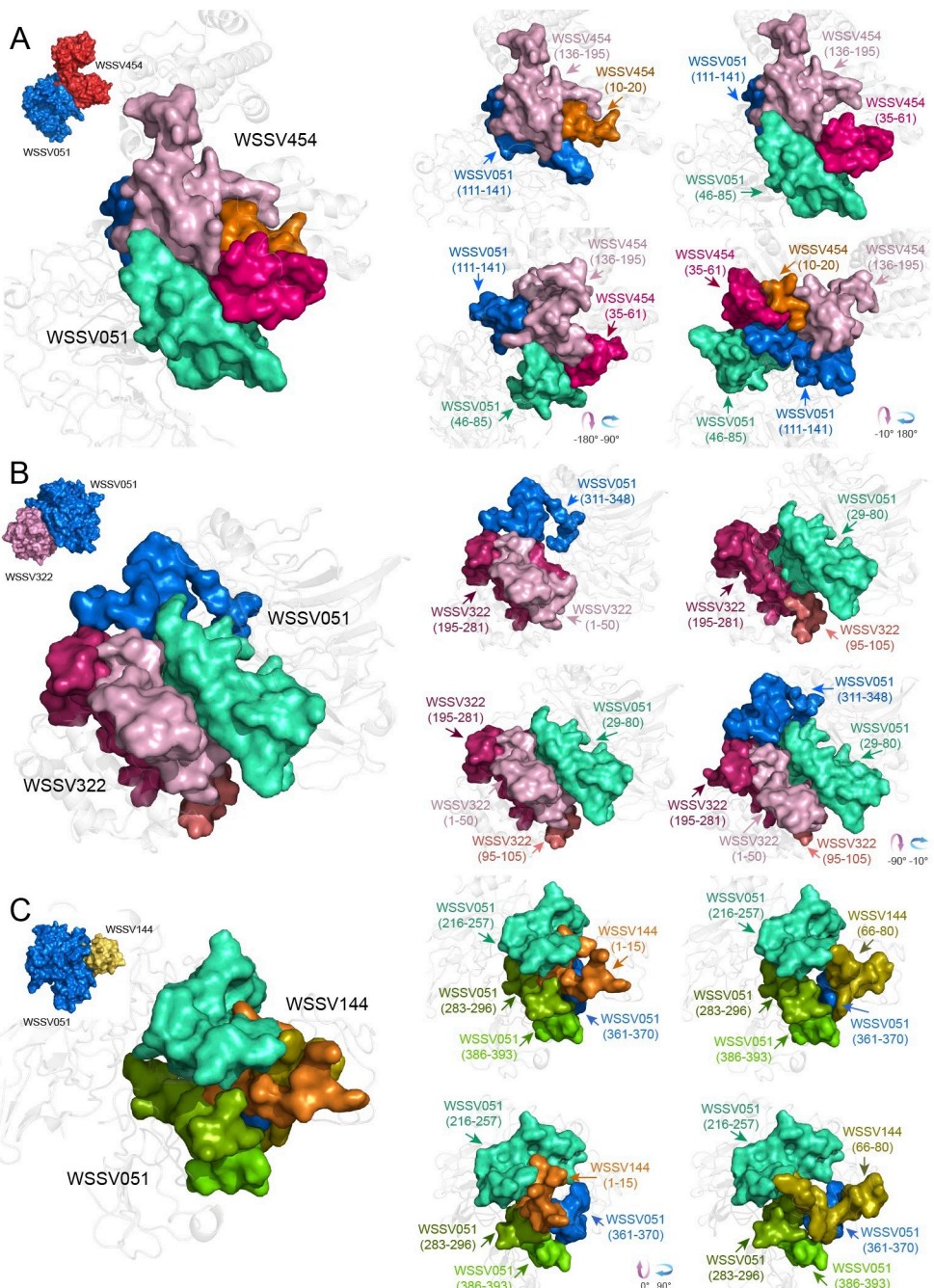

**Fig 2. Interaction complexes of WSSV051 and its binding partners.** Molecular docking of WSSV051 (blue-marine), assigned as a receptor, with WSSV454 (cherry-red), WSSV322 (chocky-pink), and WSSV144 (golden-yellow), assigned as ligand proteins, was performed on the ClusPro 2.0 server. The first ranking of the simulated complex was selected for determination of protein binding sites. The right panel shows the details of different binding sites between WSSV051 (amino acid residues 46–85 and 111–141) and WSSV454 (amino acid residues 10–20, 35–61, and 136–195) (A), WSSV051 (amino acid residues 29–80 and 311–348) and WSSV322 (amino acid residues 1–50, 95–105, and 195–281) (B), and WSSV051 (amino acid residues 216–257, 283–296, 361–370, and 386–393) and WSSV144 (amino acid residues 1–15 and 66–80) (C).

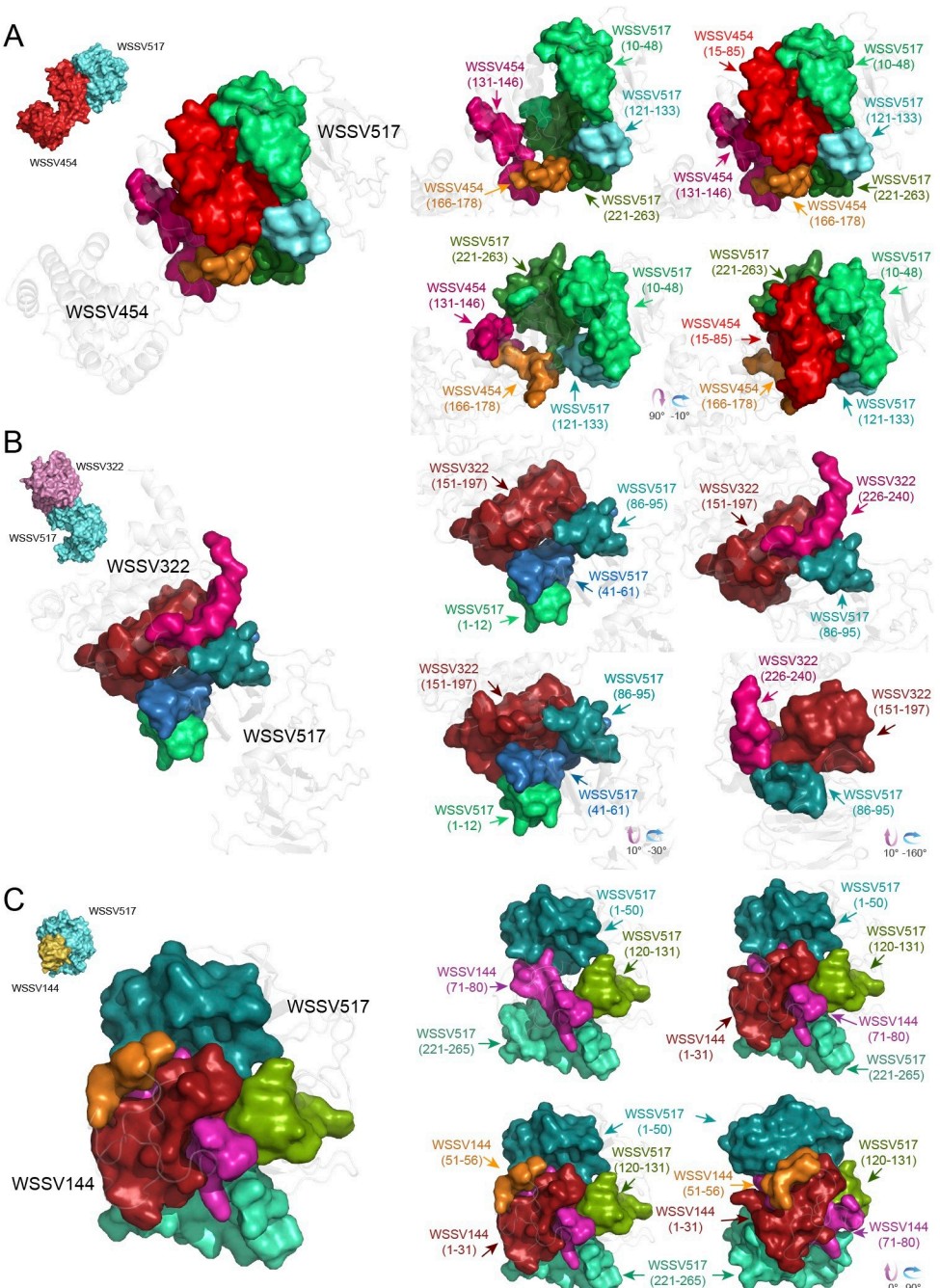

**Fig 3. Interaction complexes of the WSSV517 hub protein and its binding partners.** The 3D model of WSSV517 (turquoise), assigned as a receptor, was molecularly docked with WSSV454 (cherry-red), WSSV322 (chocky-pink), and WSSV144 (golden-yellow), assigned as ligand proteins. The right panel shows the details of the binding sites between WSSV517 (amino acid residues 10–48, 121–133, and 221–263) and WSSV454 (amino acid residues 15–85, 131–146, and 166–178) (A), WSSV517 (amino acid residues 1–12, 41–61, and 86–95) and WSSV322 (amino acid residues 151–197 and 226–240) (B), and WSSV517 (amino acid residues 1–50, 120–131, and 221–265) and WSSV144 (amino acid residues 1–31, 51–56, and 71–80) (C).

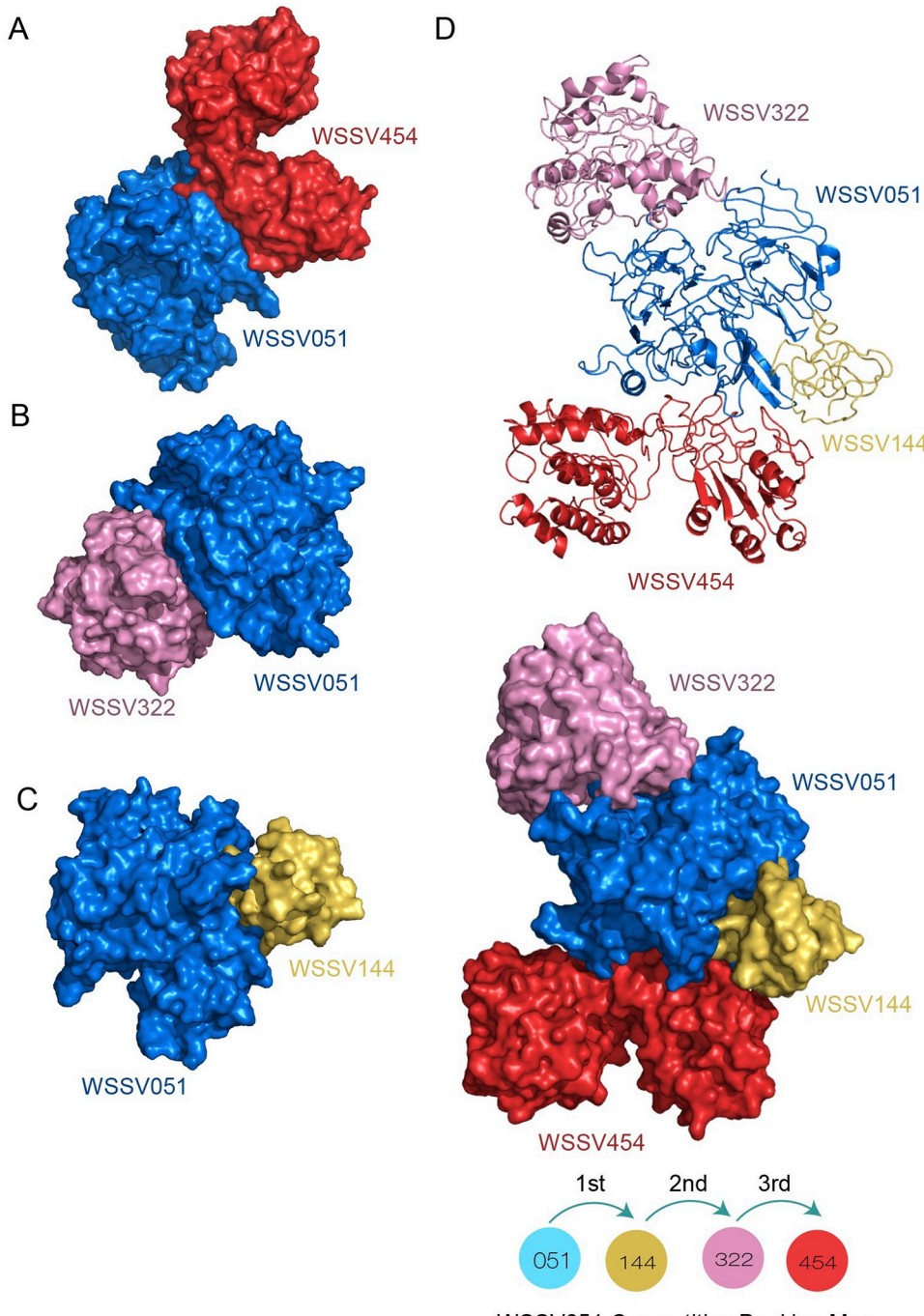

**Fig 4. Competitive docking simulation of WSSV051 with its binding partners.** A competitive docking model of WSSV051 (marine blue) was simulated to bind with ligand proteins in the following order: WSSV144 (golden yellow) →WSSV322 (chocky pink) →WSSV454 (cherry red) (A–C). The supercomplex of WSSV051 with multiple partners is demonstrated as cartoon structures (D).

binding scores of WSSV517:WSSV322 (-783 kcal/mol of center binding energy and -799.30 kcal/mol of lowest energy) are lower than those of the WSSV517:WSSV144 complex (Table 2). Furthermore, a large perspective of amino acid side chain interactions between WSSV hubs and interacting partners is also shown in S5–S10 Figs in S1 File.

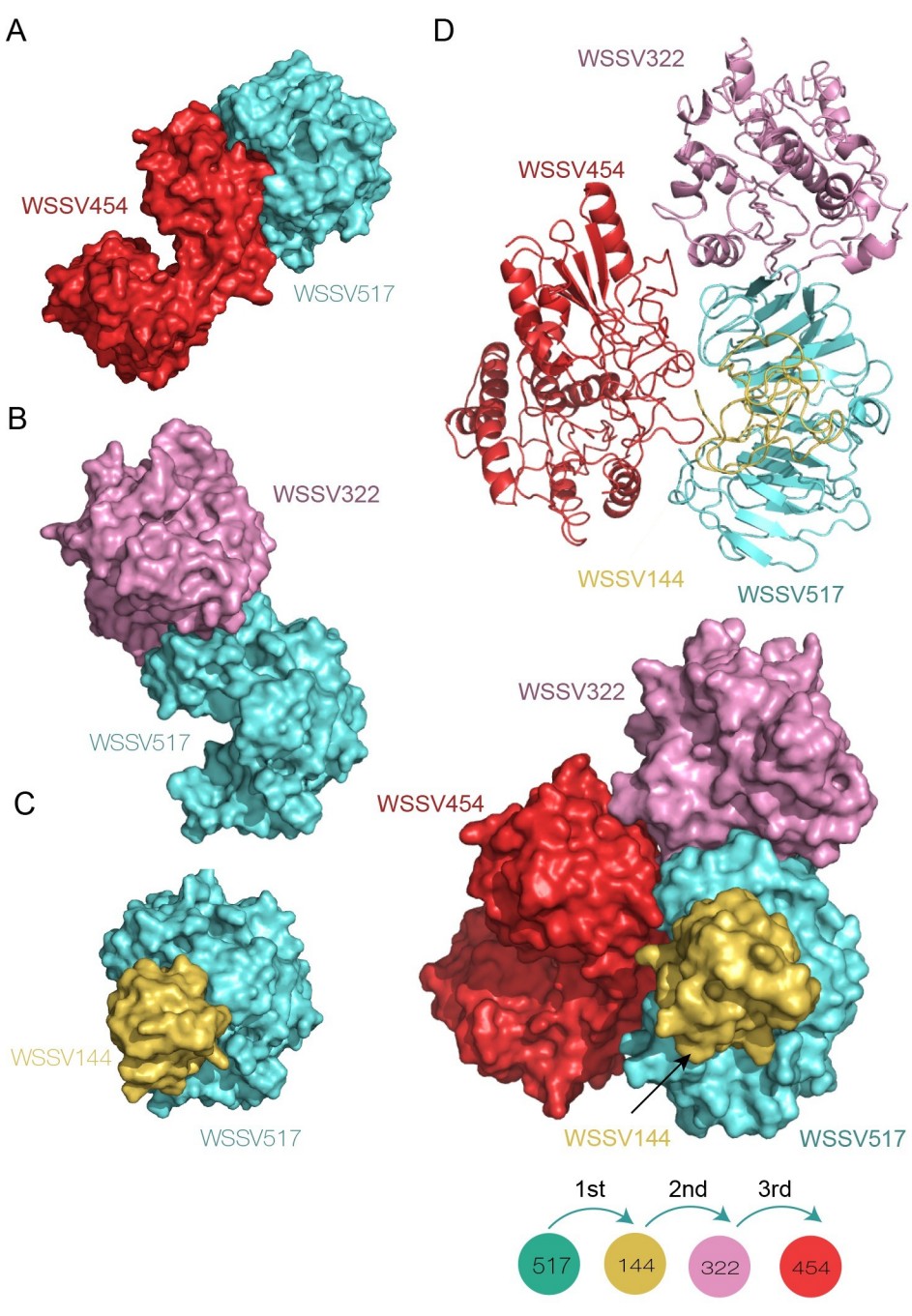

**Fig 5. Competitive docking simulation of WSSV517 with its binding partners.** The competitive map of WSSV517:
WSSV144/WSSV322/WSSV454 was simulated (A–C). WSSV517 (marine blue) was simulated to bind with its partners
in the order WSSV144 (golden yellow) →WSSV322 (chocky pink) →WSSV454 (cherry red). The supercomplex of
WSSV517 with multiple partners is presented as cartoon structures (D).

In the competitive docking simulation, the superinteraction complex of WSSV051 with
multiple interacting proteins (WSSV051[Receptor]: WSSV454[1st-Ligand]/WSS322[2nd-Ligand]/
WSSV144[3rd-Ligand]) is shown in Fig 4, suggesting that the WSSV051 protein exhibits a flexible

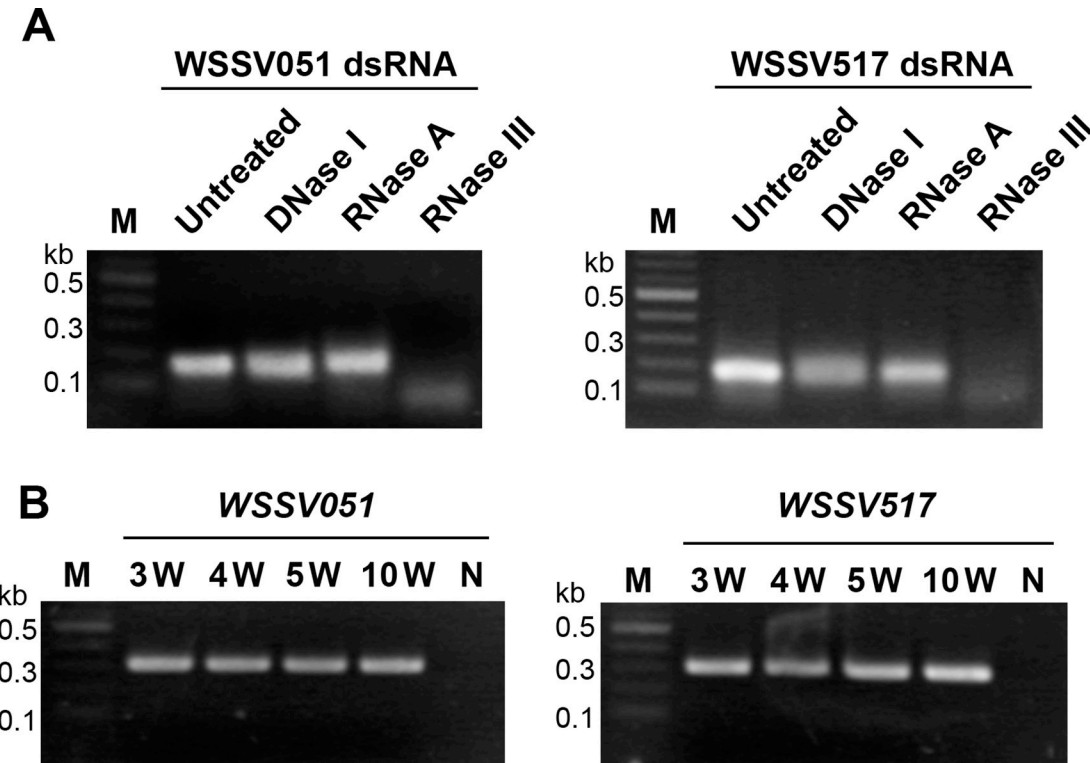

**Fig 6. Integrity and stability of WSSV hub dsRNA.** (A) WSSV051 and WSSV517 dsRNA extracted from bacterial cultures were characterized by nuclease treatments using DNase I, RNase A and RNase III. The digested products were observed by gel electrophoresis and compared with the untreated dsRNA and 2-log DNA ladders (M). (B) At 3, 4, 5 and 10 weeks after feed preparation, the presence of dsRNAs in feed was investigated by RT–PCR using RNA extracted from feed mixed with bacterial cultures as template and specific primers to WSSV051 and WSSV517 genes. The negative control (N) was conducted using distilled water replaced with template.

conformation, as seen in hub proteins. The ordering maps of all interacting proteins with WSSV051 and different superinteraction complexes are shown in S11 Fig in S1 File. Similarly, the competitive simulation results showed that WSSV517 operates as a hub protein at the center of the WSSV superstructure, with WSSV454, WSSV322, and WSSV144 acting as binding partners around the hub (Fig 5). The ordering maps of all attached molecules with WSSV517 and various superinteraction complexes are further shown in S12 Fig in S1 File. In addition, superstructural analysis of WSSV hubs suggested that WSSV051 has more possible conformations that arrange hub proteins at the center of the superstructure than WSSV517. This could be supported by the binding energy results of the WSSV051 and WSSV517 competitive docking simulation, as shown in S4 Table in S2 File. In conclusion, the WSSV complexes described here are useful for future research, such as finding new drugs or compounds that can disrupt WSSV protein complexes.

## Investigating the potential use of WSSV hubs through RNA interference

The dsRNA specific to WSSV hub genes was produced from bacterial culture, with yields of 17.08 ± 2.91 mg/l for WSSV051 dsRNA and 20.12 ± 2.61 mg/l for WSSV517 dsRNA. The integrity of WSSV051 dsRNA and WSSV517 dsRNA, synthesized in bacterial cells, was evaluated through nuclease digestion. The results confirmed their dsRNA nature as they were completely digested with RNase III but remained unaffected by DNase I and RNase A (Fig 6A).

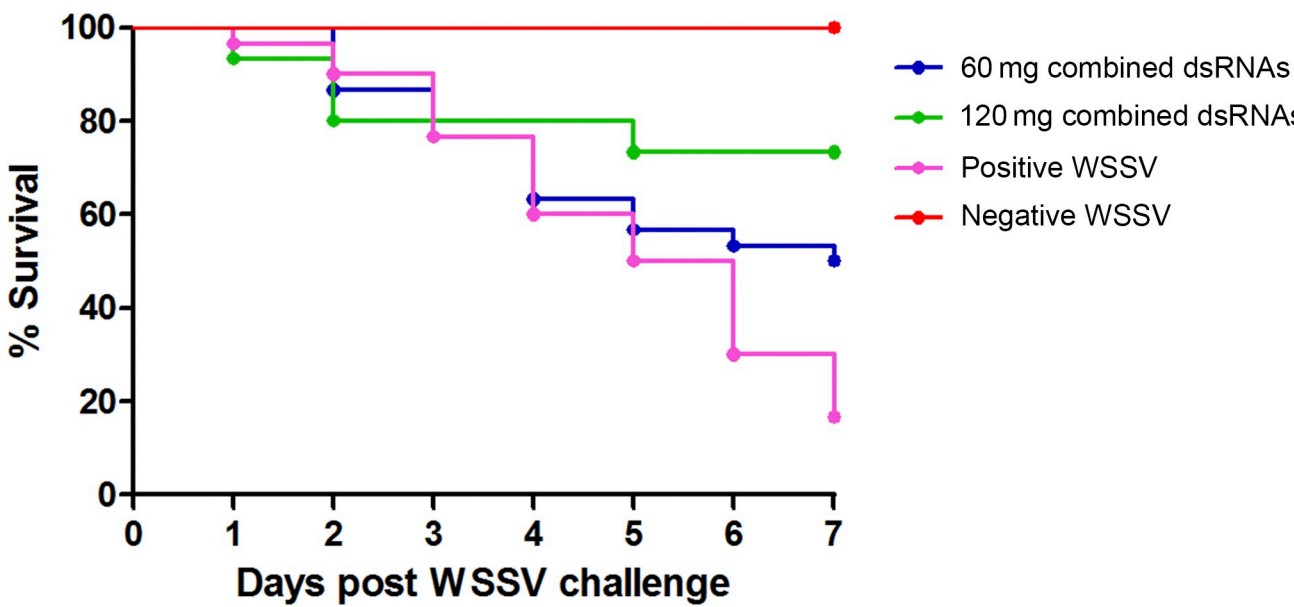

**Fig 7. Protection against WSSV infection by combined hub dsRNA mixed feed.** The *P. monodon* shrimp in each group (n=30) were fed with feed mixed with bacterial cells expressing WSSV051 and WSSV517 dsRNA (60 and 120 mg dsRNA/kg feed). After feeding for 3 days, the shrimp were challenged with WSSV. Control groups were fed with feed mixed with non-expressing dsRNA bacteria and challenged with WSSV (positive control) or with NaCl (negative control). During the experimental periods, the shrimp were continuously fed dsRNA mixed feed, and their survival rates were monitored for 7 days.

The bacterial cells expressing WSSV hub dsRNA were combined and mixed with shrimp feed for oral delivery of the dsRNAs. The stability of both dsRNAs in the shrimp feed was determined by the presence of amplified bands in RT-PCR assays conducted at 3, 4, 5 and 10 weeks post-storage at room temperature (Fig 6B).

## Protection of shrimp mortality by orally delivered combined WSSV hub dsRNA

The survival graph shown in Fig 7 illustrated that *P. monodon* fed either 60 mg or 120 mg of combined dsRNAs had a higher survival rate than those fed non-expressed dsRNA bacterial cells as a positive control. At the end of the experiment, the survival rates of the 60 mg and 120 mg combined dsRNA dosage groups were 50% and 73%, respectively, while the control group received non-expressed dsRNA and showed a survival rate of 17%. Although the study lacked statistical analysis, the findings demonstrate that oral delivery of a double-dose combined WSSV hub dsRNA could effectively inhibit viral genes via RNAi, thereby preventing shrimp mortality.

## Reduction in WSSV replication by orally delivered combined WSSV hub dsRNA

The gene knockdown efficiency of the combined hub dsRNA was examined by monitoring the transcripts of WSSV051 and WSSV517 using RT-PCR in shrimp at 6 days post WSSV infection. In shrimp receiving 60 mg of combined dsRNA feed, the WSSV517 transcript was clearly reduced, whereas the WSSV051 transcript was not well suppressed, as observed by gel electrophoresis. In 120 mg of combined dsRNA-treated shrimp, both WSSV051 and WSSV517 transcripts were not detected (Fig 8A). These data confirmed that high doses of combined dsRNA

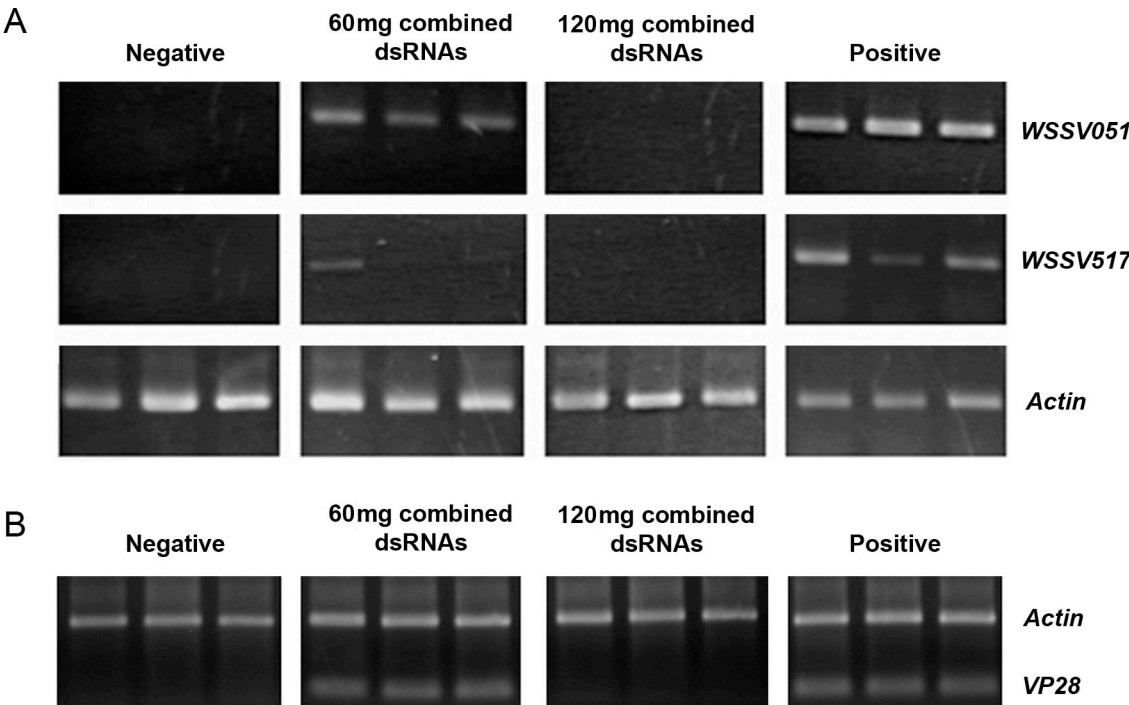

**Fig 8. Efficiency of WSSV hub gene silencing by combined dsRNA mixed feed.** The shrimp fed with feed mixed with bacterial cells expressing WSSV051 and WSSV517 dsRNA (60 and 120 mg dsRNA/kg feed) for 3 days were challenged with WSSV. Control groups fed with feed mixed with non-expressing dsRNA bacteria were challenged with WSSV (positive control) or with NaCl (negative control). (A) At 6 days post-WSSV infection, WSSV051 and WSSV517 transcripts in shrimp gill tissues were investigated by RT–PCR. Each lane represents an individual shrimp in each treatment. Shrimp actin transcripts served as the internal control. (B) The viral loads from the shrimp specimens were monitored by PCR amplification of VP28. Amplified products of shrimp actin and VP28, separately amplified from the same shrimp sample, were loaded into the same well for visual comparison.

were effective in suppressing WSSV051 and WSSV517 gene expression using an RNAi method. The effect of knocking down WSSV hub genes on WSSV infection was further investigated using PCR detection of VP28, which encodes a WSSV envelope protein and could indicate that the viral protein assembles completely as viral particles. VP28 was amplified and detected, as PCR products gave rise to the band in 60 mg of combined dsRNA feed-treated shrimp. On the other hand, no DNA band corresponding to VP28 was found in 120 mg of combined dsRNA-treated shrimp (Fig 8B), suggesting that WSSV replication was decreased by oral delivery of high doses of WSSV hub dsRNA.

## Discussion

In the present study, computational-based simulations were performed to create model sampling and illustrate WSSV-WSSV protein interactions. Primary and secondary structure analysis of WSSV051 and WSSV517 and their mutual binding partners found prominent characteristics that contribute to the hub proteins including low-complexity regions (LCRs), disordered protein binding region, and linear protein motifs. LCRs are regions of a protein that have non-globular structures and are flexible in nature [32]. These regions play a role in the creation of stable helices [33]. Additionally, protein sequences may contain LCRs that play a role in regulating protein interactions and immune evasion mechanisms [34–37]. LCRs are typically found in intrinsically disordered regions (IDRs), which are characterized by their inability to form secondary or tertiary structures [38–40]. In viruses, viral protein disorder

regions provide a purpose connected to the flexibility of conformation structures and associated mechanisms [41]. One example of hub proteins called Huntingtin (HTT) that have hundreds of protein interactors and mutations in the HTT gene are associated with Huntington's disease is well described for a variety of LCRs that contribute to different modes of interaction [40]. Therefore, the results in the present study suggest that the identified LCR regions of WSSV051 and WSS517, along with their interactors, may induce flexibility in the conformation of bound molecules, which can be regarded as a characteristic of WSSV hubs. In addition to the unique features of WSSV hubs, the pore-lining regions present in WSSV051 and WSSV517 may participate in the interaction with cell membranes and the regulation of viral protein transportation. This is because the pore-lining regions have crucial functions in facilitating the movement of molecules across the cell's bilayer [42, 43].

Repeated motifs within WSSV hubs and their interacting proteins appear to be involved in protein-protein interactions. This finding is supported by the docking model of WSSV hubs and their interactors, revealing crucial binding sites within protein complexes. Notably, two distinct protein motifs, namely "S-G-x(2)-S-x(2)-T-x(2)-N-S" and "N-S-x(1,2)-V-G-x-L-x(5)-D" are present in both WSSV051 and WSSV517; however, it seemed that these motifs indicated a more favorable interaction by WSSV517. For instance, the "S-G-x(2)-S-x(2)-T-x(2)-N-S" in WSSV517 at Ser32-Ser43 is located at the interaction sites with WSSV144 and WSSV454, while the "N-S-x(1,2)-V-G-x-L-x(5)-D" motif in WSSV517 at Asn46-Asp58 contributes to the interaction with WSSV322. Furthermore, the hub-binding proteins WSSV322 (at Ser225–Pro235) and WSSV454 (at Ser67–Pro77) exhibited a highly similar sequence pattern, "S-S-x(5)-S-x(2)-P", facilitating interaction with WSSV517. Short linear motifs (SLiMs) are a fundamental category of protein interaction modules, alongside globular domains and intrinsically disordered domains [44]. In recent times, a category of hub proteins has been more precisely characterized as linear motif-binding hub proteins (LMB-hubs). These hubs specifically bind to an 8-10 amino acid residue consensus motif on a partner protein and can serve either an enzymatic function (interacting with consensus motifs temporarily) or a structural function (stabilizing partner-protein complexes) [45]. Unlike WSSV517, the repetitive patterns in WSSV051 and their interacting partners are unlikely to be involved in protein-protein interaction, as the repeat patterns are not present in the interaction locations. Nonetheless the docking of WSSV051 and its interactors provided evidence that WSSV051 functions as a hub due to its favorable binding energy scores.

Within a PPI network, hubs can be categorized as either single-interface hubs, also known as date hubs, or multi-interface hubs, referred to as party hubs. Single-interface hubs possess a maximum of two interaction interfaces that are utilized by many partners. On the other hand, multi-interface hubs possess more than two interaction interfaces and have the ability to interact with several partners concurrently [46]. The presence of several interaction sites in WSSV hubs and their corresponding binding partners suggests that WSSV hubs may possess a characteristic known as multi-interface hubs. This was supported by the WSSV-WSSV protein competitive docking simulation, in which WSSV051 and WSSV517 demonstrated the flexible structure that may be necessary for assembly with different partners. Notably, WSSV051 has more possible conformations that arrange hub proteins at the center of the superstructure than WSSV517. Nevertheless, it is imperative to conduct further research on the co-expression of WSSV hubs and their partners. This is because party hubs exhibit gene expression patterns that are strongly associated with those of their partners [46]. Additionally, competitive binding assays and mutagenesis of these WSSV proteins could help validate interaction specificity and provide insights into their binding kinetics. Understanding this more dynamic view of the interaction through WSSV hubs should shed light on the functional significance of the interaction and WSSV biology. Although, the functional role of the binding between WSSV hubs and

**Table 3. Oral delivery of different dsRNA formulations against virus infection in shrimp.**

| dsRNA formulation | Amount of dsRNA or cells expressing dsRNA | Virus (target gene) | Specie | Protection effect | Reference |
|---|---|---|---|---|---|
| Feed formulated with bacteria expressing dsRNA | 60 mg and 120 mg dsRNA/kg feed | WSSV (WSSV051 and WSSV517) | *P. monodon* | 50-73% survival | This study |
| | 6 mg and 12 mg dsRNA/kg feed | WSSV (WSSV051 and VP28) | *P. vannamei* | 40-75% survival | [13] |
| | $5.0 \times 10^7$ cells/g feed | WSSV (VP28) | *P. monodon* | 68% survival | [47] |
| | 6 mg and 12 mg dsRNA/kg feed | LSNV (RdRp) | *P. monodon* | 20-60% reduction of LSNV | [49] |
| | $5.0 \times 10^9$ cells/g feed | MrNV (Capsid and B2) XSV (Capsid) | *M. rosenbergii* | 45-80% survival | [51] |
| | $4.0 \times 10^{10}$ cell/g feed | GAV (ORF1a/b) | *P. monodon* | No protection | [59] |
| Feed formulated with dsRNA-chitosan nanoparticles | 35 µg dsRNA/100 µl chitosan solution | WSSV (VP28) | *P. monodon* | 37% survival | [47] |
| Bacterial expressed dsRNA embedded in agar | $1.5 \times 10^{11}$ cells/ml agar solution | YHV (Protease) | *P. vannamei* | 70% survival | [48] |
| *Artemia* enriched with bacteria expressing dsRNA | $4.3 \times 10^{11}$ cells/$2 \times 10^6$ *Artemia* | LSNV (RdRP) | *P. monodon* | 55% reduction of LSNV | [50] |
| Feed formulated with microalgae expressing dsRNA | $1.0 \times 10^8$ cells/g feed | YHV (RdRP) | *P. vannamei* | 22% survival | [60] |
| Microalgae expressing dsRNA without antibiotic supplementation | $5 \times 10^5$ algal cells/ml seawater | YHV (RdRP) | *P. vannamei* | 50% survival | [61] |

their partners in the WSSV life cycle is still unknown in our current study, our initial findings on WSSV-WSSV docking simulations should serve as guidelines for future research, such as the development of antiviral peptide-targeted designs.

Regarding the disease control strategies in aquaculture, RNAi has emerged as a highly promising technique for controlling viral infections in shrimp aquaculture. However, the administration of siRNA or dsRNA to trigger RNAi in shrimp through injection is not feasible in the situation of shrimp farming. Consequently, the oral administration of siRNA or dsRNA is a preferable method. Oral delivery of dsRNA has been shown to inhibit shrimp viruses such as WSSV, yellow head virus (YHV), Laem-Singh virus (LSNV), *Macrobrachium rosenbergii* nodavirus (MrNV), and an extra small virus (XSV) [47–51]. Most of viral suppression targets are structural and enzymatic genes that are necessary for viral replication. For example, WSSV knockdown targets include VP28, VP26, VP37, and rr2, whereas YHV targets include gp116, gp64, RdRp, and rr2 [52]. The utilization of many target genes has also enhanced the effectiveness of viral suppression. According to Sanitt et al. [48], administering a combination of dsRNAs targeting shrimp and viral genes (namely dsRab7 and dsYHV) orally can enhance protection against YHV compared to using dsYHV alone. Thammasorn et al. [13] observed that the oral administration of multitarget dsRNA for the WSSV051 hub gene and VP28 showed inconsistent levels of protection against WSSV infection. Hence, our research proposed the utilization of new multitargets, WSSV051 and WSSV517, for suppression by means of oral delivery of their respective dsRNAs. Apart from shrimp feed formulation, various materials could be applied in the delivery of dsRNA, such as chitosan, microalgae and *Artemia*. The different oral delivery methods of dsRNA against various shrimp viruses are summarized in Table 3.

It has been clearly proven that shrimp feeding with bacteria expressing specific dsRNA can induce the systemic dissemination of RNAi signals from the hepatopancreas through the hemolymph and gills to silence shrimp genes Rab7 and STAT [53, 54]. The process by which dsRNA is taken up into hepatopancreatic cells is described by the mechanism of clathrin-

mediated endocytosis [54, 55]. While the process in which RNAi signals are transported from hepatopancreatic cells to shrimp hemolymph is not yet understood, it has been observed that the RNAi signal, specifically long dsRNA, is then taken in by the gills via the function of the systemic RNA interference deficient 1 (SID-1) protein [54]. This discovery could provide support for the current study that the ingested dsRNA specific to WSSV051 and WSSV517 can suppress WSSV hub transcripts in gill tissues. However, it was found that the efficiency of the inhibitory effect against WSSV was low to some extent, especially knocking down WSSV051. A similar observation was noticed in the previous investigation, where the knockdown of WSSV051 by the injection of WSSV051 dsRNA resulted in lower suppression activity compared to the injection of WSSV517 dsRNA [7]. The varied knockdown efficiency led to different protective effects, which were speculated to be associated with the functionality of the gene-encoded proteins and their roles throughout the viral life cycle [56]. Given this, shrimp feed containing only WSSV517 dsRNA might be a promising option for future applications. In addition to knockdown targets, the concentration of dsRNA in the diet is the most critical factor in determining gene suppression efficacy. According to Sanitt et al. [48], oral administration of dsRNA requires 400-800 times higher concentrations than injection to prevent YHV infection because dsRNA can be degraded by digestive enzymes in the shrimp hepatopancreas and stomach. Although our study did not determine the exact amount of dsRNA in shrimp feed due to the unavailability of the RT-qPCR method for dsRNA quantification in our laboratory, we investigated the potential degradation of dsRNA within 3–10 weeks after preparation. The stability of dsRNA was confirmed by the presence of RT-PCR products for both WSSV051 and WSSV517, indicating that dsRNA in shrimp feed remains present and functional when fed to shrimp. Additionally, it was shown that dsRNA could be maintained for up to 7 months after feed preparation, though some degree of degradation was observed [13]. Our investigations also revealed that increasing the dose of WSSV051 and WSSV517 dsRNA in shrimp diets resulted in a significant enhancement in gene suppression, thereby leading to increased survival rates in WSSV-infected shrimp. This finding is supported by PCR amplification of VP28, which detected VP28 in shrimp fed with 60 mg of combined dsRNA feed but not in those fed with 120 mg of combined dsRNA feed. Additional assays, such as Western blotting to detect WSSV proteins like VP28, as well as histopathology, would be beneficial in further confirming the extent of WSSV infection. Despite the numerous benefits of the bacterial-expressing dsRNA system, such as its simplicity, high productivity, and cost-effectiveness, there are apprehensions over the use of *E. coli* in shrimp and its potential environmental impacts. Therefore, searching for alternative hosts such as *Corynebacteirum glutamicum*, *Chlamydomonas reinhardtii*, and *Saccharomyces cerevisiae*, known as GRAS (generally recognized as safe) microorganisms [57], for dsRNA production of such new viral targets (i.e., WSSV hubs), is a challenge for controlling WSSV in shrimp aquaculture.

## Conclusion

Using advanced computational simulation, we first exhibited 3D modeling of lacking templates in WSSV hub proteins and binding partners. The possible complexes of WSSV hub proteins and partners with identifiable substantial interacting surfaces were then visualized using precise molecular docking methods. The superinteraction complex of WSSV051 or WSSV517 with numerous interacting partners was also revealed by competitive docking simulation, indicating the hub property of these two WSSV proteins. The WSSV hub complex described here serves as a starting point for more research into new drugs or inhibitory compounds that can disrupt the WSSV complex. In addition to structural analysis of WSSV hubs, this study used RNAi to elucidate the importance of WSSV051 and WSSV517 knockdown through oral

delivery of combined dsRNAs. The results showed that double-dose combined dsRNAs expressed in *E. coli* could help maximize the efficacy of the antiviral response in *P. monodon* to WSSV because they led to the lowest shrimp mortality rate and diminished WSSV replication. Although *E. coli* used to express dsRNA may not be the best host organism due to food safety concerns about its antibiotic resistance, this study provided the potential application of hub dsRNAs to facilitate protection against WSSV, especially in farmed shrimp. Further insight is required to investigate alternative approaches, including a suitable feeding method and selection of target genes for optimization on an industrial scale. Eventually, future studies will be considerably beneficial to the economic growth resulting from the antiviral strategy in shrimp culture.

## Supporting information

**S1 File. Supplementary figs 1–12.**
(PDF)

**S2 File. Supplementary tables 1–4.**
(XLSX)

**S1 Raw images.**
(PDF)

## Author Contributions

**Conceptualization:** Tanate Panrat, Saengchan Senapin, Pakkakul Sangsuriya.

**Data curation:** Tanate Panrat, Saengchan Senapin, Pakkakul Sangsuriya.

**Formal analysis:** Tanate Panrat, Pakkakul Sangsuriya.

**Funding acquisition:** Tanate Panrat, Amornrat Phongdara, Pakkakul Sangsuriya.

**Investigation:** Tanate Panrat, Kitti Wuthisathid, Watcharachai Meemetta, Kornsunee Phiwsaiya, Pakkakul Sangsuriya.

**Methodology:** Tanate Panrat, Saengchan Senapin, Pakkakul Sangsuriya.

**Project administration:** Saengchan Senapin.

**Resources:** Rapeepun Vanichviriyakit.

**Supervision:** Amornrat Phongdara, Saengchan Senapin.

**Writing – original draft:** Tanate Panrat, Kitti Wuthisathid, Pakkakul Sangsuriya.

**Writing – review & editing:** Tanate Panrat, Saengchan Senapin, Pakkakul Sangsuriya.

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
