## [Decision Letter · Decision Letter 0]

10 May 2024

PONE-D-24-15796Structural Modelling and Preventive Strategy Targeting of WSSV Hub Proteins to Combat Viral Infection in Shrimp Penaeus monodonPLOS ONE

Dear Dr. Sangsuriya,

Thank you for submitting your manuscript to PLOS ONE. After careful consideration, we feel that it has merit but does not fully meet PLOS ONE’s publication criteria as it currently stands. Therefore, we invite you to submit a revised version of the manuscript that addresses the points raised during the review process.

We look forward to receiving your revised manuscript.

Kind regards,

Kenneth Söderhäll

Academic Editor

PLOS ONE

Journal Requirements:

"This research project is supported by Mahidol University (Fundamental Fund: fiscal year 2023 by the National Science Research and Innovation Fund (NSRF), Grant no. FF-056/2566. TP and AP acknowledge the funding support from the NSRF and Prince of Songkla University (Grant no. UIC6601175S)."

"Mahidol University (Fundamental Fund: fiscal year 2023 by the National Science Research and Innovation Fund (NSRF), Grant no. FF-056/2566 and the NSRF and Prince of Songkla University (Grant no. UIC6601175S)."

 "Mahidol University (Fundamental Fund: fiscal year 2023 by the National Science Research and Innovation Fund (NSRF), Grant no. FF-056/2566 and the NSRF and Prince of Songkla University (Grant no. UIC6601175S)."     

Reviewers' comments:

Reviewer's Responses to Questions

**Comments to the Author**

1. Is the manuscript technically sound, and do the data support the conclusions?

Reviewer #1: Yes

Reviewer #2: Yes

2. Has the statistical analysis been performed appropriately and rigorously? 

Reviewer #1: Yes

Reviewer #2: Yes

3. Have the authors made all data underlying the findings in their manuscript fully available?

Reviewer #1: Yes

Reviewer #2: Yes

4. Is the manuscript presented in an intelligible fashion and written in standard English?

Reviewer #1: Yes

Reviewer #2: Yes

5. Review Comments to the Author

Reviewer #1: The manuscript presents a wealth of bioinformatics analysis results, but the experimental section is comparatively weak. There are several issues that need to be addressed for improvement.1. In the “preparation of shrimp feed containing WSSV hub dsRNA” section, it is crucial to clarify how the authors ensure that each kilogram of feed contains the specified amounts of dsRNA (60 mg or 120 mg). Additionally, the manuscript should address the yield of dsRNA production from bacteria when mixed with feed, its stability, and the potential for degradation within 3-10 weeks post-feed preparation. Furthermore, while the authors have successfully detected the presence of dsRNA through RT-PCR and agarose gel electrophoresis, they should also focus on determining its content accurately.2. Why did the authors choose the intramuscular injection method for virus infection in the LD50 experiment instead of the immersion infection method? Given that immersion infection mirrors aquaculture conditions more closely. Is there any death due to mechanical damage from intramuscular injection? Fig.7 shows shrimp death on the first day of injection. And the manuscript should provide data on the virus titer.3. Have functional studies been conducted on WSSV517 and WSSV510? As hub proteins, they are expected to play crucial roles in various physiological processes during virus infection. Please provide additional information in the introduction.4. The manuscript should elaborate on the physiological processes in which the ligands WSSV144, WSSV454, and WSSV322 are involved and the regulatory functions they perform.5. While the protein interactions presented in the manuscript are simulated by computer, it is recommended to provide Co-IP validation of these interactions. Moreover, constructing mutants to validate specific protein interaction sites would enhance the rigor of study.6. Both of WSSV051 and WSSV517 can bind to WSSV144, WSSV454, and WSSV322. The potential competitive relationships between WSSV051 and WSSV517, should be explored further.7. There are two Fig. 7 in the manuscript. They are different that need to be addressed for clarity.8.In the second Fig. 7, it is essential to label the feeding amounts of WSSV051 dsRNA and WSSV517 dsRNA for completeness

. 9.The results in Fig. 7 and Fig. 8 indicate that the antiviral effect of WSSV517 dsRNA surpasses that of WSSV051 dsRNA, and feeding shrimp with feed containing WSSV517 dsRNA results in higher survival rates than feeding with a mixture feed of WSSV051 dsRNA and WSSV517 dsRNA. Given these findings, have the authors considered feeding only with feed containing WSSV517 dsRNA rather than a mixture of WSSV051 dsRNA and WSSV517 dsRNA?

Reviewer #2: This MS describes the use of computational methods to derive the 3D structures of WSSV051 and WSSV517 viral proteins, and identify their interacting partners. In addition, animal experiment was performed to assess the protective effects of the combined WSSV051 and WSSV517 dsRNAs against WSSV infection in P. monodon. The dsRNA incorporated feed shows stable expression of the viral genes after prolonged storage. This is an interesting study that explores the structural and functional aspects of the viral proteins and the use of RNAi could have a positive impact on shrimp disease management. The MS is well organized; however, the following suggestions would improve the manuscript.

Specific Comments:

1. In the material and methods (Lines 136-137) could the authors explain why the WSSV517 plasmid was re-constructed for this study whereas the WSSV051 plasmid was used from previous studies?

2. For dsRNA incorporated shrimp feed preparation, what was the rationale for choosing 60 mg and 120 mg dsRNA concentrations?

3. Lines 170-171, the primer information for Xba1-WSSV051-RNAi-F/ Xba1-WSSV051-RNAi-R3 is not included in Table 1.

4. In Line 180, how many shrimp were used for each experimental group?

5. In animal experiment, the number of samples (n=3) collected at a single timepoint seems to be inadequate for testing the efficiency of the combined WSSV hub dsRNA. Could the authors explain why they could not include more time points or use more samples for testing?

6. For the molecular docking studies, the WSSV144, WSSV322 and WSSV454 were used as interacting partners. Is there any specific reason for choosing these 3 genes, please explain and include this information in the Discussion section.

7. For detecting the WSSV replication, the PCR amplification of VP28 seems insufficient and I suggest the authors could have performed western blot to check the VP28 protein or a H&E staining to substantiate the PCR results.

8. In Table 3, I suggest the authors to include a column on dsRNA dosage as it would be useful for the readers.

9. In Table 3, the last row mentions “without antibiotic selection” but as per the publication it is “without antibiotic supplementation”. The authors are requested to pay attention to such details.

10. For Fig 6B, it would be interesting to see the expression of the WSSV051 and WSSV517 genes in freshly prepared feed along with the stored dsRNA feed.

11. In Fig. 7, was the difference in survival % between 60 mg and 120 mg statistically significant? If so, please include this information in the Fig. 7 and in the Results section.

12. In the Figure legends of Fig 7 and 8, please include the information about the positive and negative controls used.

13. The authors are requested to check for spelling and typo errors in the MS. For example: In Fig. 8A&B, the PCR results title is mentioned as “combinded”

14. In Fig. 8, figure legend (Lines 743-744), the authors mention “Shrimp actin and VP28 were separately amplified in each reaction but loaded in the same well.” Could the authors explain the reason for doing so?

6. PLOS authors have the option to publish the peer review history of their article (what does this mean?). If published, this will include your full peer review and any attached files.

Reviewer #1: No

Reviewer #2: No

---

## [Author Response · Author response to Decision Letter 0]

14 Jul 2024

Editor: Thank you for submitting your manuscript to PLOS ONE. After careful consideration, we feel that it has merit but does not fully meet PLOS ONE’s publication criteria as it currently stands. Therefore, we invite you to submit a revised version of the manuscript that addresses the points raised during the review process.

Response: Thank you for handling our manuscript and granting us an extension for the revision. We appreciate the time and effort you and the reviewers have invested in evaluating our work. We have carefully considered each point raised and have made the necessary revisions to address them point-by-point. Please find below our detailed response to the reviewers' comments: 

Reviewer #1: The manuscript presents a wealth of bioinformatics analysis results, but the experimental section is comparatively weak. There are several issues that need to be addressed for improvement.

Thank you for your thoughtful review and for acknowledging the strengths of the bioinformatics analysis in our manuscript. We greatly appreciate your insightful comments on the experimental section, and we have made the following revisions to address the issues you raised.

1. In the “preparation of shrimp feed containing WSSV hub dsRNA” section, it is crucial to clarify how the authors ensure that each kilogram of feed contains the specified amounts of dsRNA (60 mg or 120 mg). Additionally, the manuscript should address the yield of dsRNA production from bacteria when mixed with feed, its stability, and the potential for degradation within 3-10 weeks post-feed preparation. Furthermore, while the authors have successfully detected the presence of dsRNA through RT-PCR and agarose gel electrophoresis, they should also focus on determining its content accurately.

Response 1: The yields of dsRNAs produced from E. coli are indicated in the Results section on Lines 357-358. The text reads as follows: “The dsRNA specific to WSSV hub genes was produced from bacterial culture, with yields of 17.08 ± 2.91 mg/l for WSSV051 dsRNA and 20.12 ± 2.61 mg/l for WSSV517 dsRNA.” Based on these yields, we calculated the amounts of bacterial suspension needed to achieve the desired dsRNA quantities (i.e., 60 mg or 120 mg) when mixing with shrimp feed. However, we did not determine the exact dsRNA content in the shrimp feed after preparation due to the absence of quantitative PCR methods for dsRNA measurement. This limitation is acknowledged in the revised manuscript, with the respective description provided in Lines 502-505 of the Discussion.

To test the stability of the dsRNA in the feed, we conducted RNA extractions and RT-PCR assays on samples taken at 3, 4, 5, and 10 weeks post-storage at room temperature. Details of these procedures are provided in the Materials and Methods section (Lines 187-192), the Results section (Lines 363-365), and are illustrated in Figure 6B (Lines 777-780).

The potential for dsRNA degradation within 3-10 weeks post-feed preparation is discussed in the Discussion section. The text can be found in Lines 502-509.

2. Why did the authors choose the intramuscular injection method for virus infection in the LD50 experiment instead of the immersion infection method? Given that immersion infection mirrors aquaculture conditions more closely. Is there any death due to mechanical damage from intramuscular injection? Fig.7 shows shrimp death on the first day of injection. And the manuscript should provide data on the virus titer.

Response 2: We agree with the Reviewer that the WSSV immersion challenge closely resembles a natural viral infection. However, to ensure uniform infection across the shrimp population, we employed intramuscular injection. The use of a 26-gauge needle for intramuscular injection did not result in mortality, as evidenced by the absence of deaths in the NaCl-injection control group. Shrimp mortality observed one day post-WSSV challenge occurred only in the WSSV-challenged groups, not in the NaCl-injected group, indicating that the mortality is likely attributable to viral infection. Based on the Reviewer’s comment, we have provided additional details about the shrimp injection procedure, including specifics about the needle used (Lines 208-210). Regarding the viral titer, we did not check the exact viral copy number due to some limitations in our lab, but we performed the viral challenges with various dilutions of crude viral stock (i.e., 1:100, 1:1000, and 1:1000). This prior viral challenge can also give the desired viral dose and mortality rate when we perform the experimental dsRNA feeding and WSSV challenge. 

3. Have functional studies been conducted on WSSV517 and WSSV051? As hub proteins, they are expected to play crucial roles in various physiological processes during virus infection. Please provide additional information in the introduction.

Response 3: Previous studies have first identified that WSSV051 and WSSV517 are structural proteins VP55 and an early viral protein, respectively. We later discovered that these two proteins are important hub proteins in the WSSV protein interaction map, and they may play roles in various viral biological processes related to their binding proteins’ putative functions. To clarify the functional studies of these WSSV hubs, we have already mentioned them in the Introduction section on Lines 82-100. The text reads as follows:

“Two hub proteins, WSSV051 and WSSV517, have been confirmed to interact with various binding partners using co-immunoprecipitation. WSSV051 was previously annotated as a structural protein VP55, as indicated by proteomic analysis of purified WSSV virions [8], while WSSV517 was identified as an early protein gene through DNA microarray analysis [9]. In the WSSV PPI network, mutual binding partners of these two hubs include WSSV144, WSSV322, and WSSV454. The function of WSSV144 is unknown, while WSSV322 is an anti-apoptotic protein and WSSV454 as a thymidine-thymidylate kinase protein (TK-TMK). An in vitro insect model has demonstrated that WSSV322 binds with the shrimp caspase protein and exhibits anti-apoptotic activity [10]. The homology similarity and thymidine kinase activity confirm WSSV454’s function as a TK-TMK protein [11,12]. This suggests that two hub proteins are likely involved in several biological processes related to the functions of their binding partners, including viral assembly, DNA replication, and nucleotide metabolism [7]. Although the precise interaction mechanisms of these hubs in viral biology remain uncertain, it has been demonstrated that they play crucial roles in the WSSV life cycle. This is evident from the fact that suppressing WSSV051 or WSSV517 resulted in a delay in shrimp mortality caused by WSSV infection [7,13]. However, more investigation into the complexity of WSSV hubs and their mutual interacting partners is still required to comprehend their properties, which could lead to their use as antiviral targets.”

4. The manuscript should elaborate on the physiological processes in which the ligands WSSV144, WSSV454, and WSSV322 are involved and the regulatory functions they perform.

Response 4: We have included the previous annotations of WSSV144, WSSV454, and WSSV322 as an unknown protein, a TK-TMK protein, and an anti-apoptotic protein, respectively. The information can be found in the Introduction section on lines 86-92. Although the functional binding between two hubs and these WSSV proteins is unknown, we point out these essential bindings for future applications on antiviral targets. The Discussion section on lines 454-458 also incorporates this information. The text reads as follows: “Although, the functional role of the binding between WSSV hubs and their partners in the WSSV life cycle is still unknown in our current study, our initial findings on WSSV-WSSV docking simulations should serve as guidelines for future research, such as the development of antiviral peptide-targeted designs.”. 

5. While the protein interactions presented in the manuscript are simulated by computer, it is recommended to provide Co-IP validation of these interactions. Moreover, constructing mutants to validate specific protein interaction sites would enhance the rigor of study.

 Response 5: We appreciate the Reviewer’s comments. In our previous study (Sangsuriya et al., Mol Cell Proteomics. 2014, 13: 269-282), we conducted Co-IP validation tests for the interactions of WSSV051 and WSSV517 with their binding partners. The respective description can be found in Lines 82-84.

Regarding the construction of mutants to validate specific protein interaction sites, we acknowledge this valuable suggestion. Although we did not perform this assay in the present study, we have addressed this validation study in the Discussion section, as indicated in Lines 450-454. The text reads as follows: “Additionally, competitive binding assays and mutagenesis of these WSSV proteins could help validate interaction specificity and provide insights into their binding kinetics. Understanding this more dynamic view of the interaction through WSSV hubs should shed light on the functional significance of the interaction and WSSV biology.”.

6. Both of WSSV051 and WSSV517 can bind to WSSV144, WSSV454, and WSSV322. The potential competitive relationships between WSSV051 and WSSV517, should be explored further.

Response 6: Thank you for your valuable feedback. We appreciate your suggestion to explore the potential competitive relationships between WSSV051 and WSSV517. We have included this point in the discussion on Lines 450-454, as mentioned above. 

7. There are two Fig. 7 in the manuscript. They are different that need to be addressed for clarity.

Response 7: We deeply apologize for this error. The first Fig. 7 in the manuscript is the correct one, which is the survival rate of four groups, including 60 mg combined dsRNAs, 120 mg combined dsRNAs, positive WSSV, and negative WSSV. The second Fig. 7 was not correct, resulting from our error in uploading the figure. We have now provided the correct figure in the revised manuscript.

8. In the second Fig. 7, it is essential to label the feeding amounts of WSSV051 dsRNA and WSSV517 dsRNA for completeness.

 Response 8: Thank you for your suggestion. We have now provided the correct figure with a label for dsRNA dosage in the revised manuscript.

9. The results in Fig. 7 and Fig. 8 indicate that the antiviral effect of WSSV517 dsRNA surpasses that of WSSV051 dsRNA, and feeding shrimp with feed containing WSSV517 dsRNA results in higher survival rates than feeding with a mixture feed of WSSV051 dsRNA and WSSV517 dsRNA. Given these findings, have the authors considered feeding only with feed containing WSSV517 dsRNA rather than a mixture of WSSV051 dsRNA and WSSV517 dsRNA?

 Response 9: We concur with your suggestion that feeding only WSSV517 dsRNA would be an interesting target for preventing WSSV infection. This is also supported by our previous work showing that injection of WSSV517 dsRNA provided a better protection effect than WSSV051 dsRNA (Sangsuriya et al., Mol Cell Proteomics. 2014, 13: 269–282). This information was included in the Discussion section on Lines 490-498. The text reads as follows:

“However, it was found that the efficiency of the inhibitory effect against WSSV was low to some extent, especially knocking down WSSV051. A similar observation was noticed in the previous investigation, where the knockdown of WSSV051 by the injection of WSSV051 dsRNA resulted in lower suppression activity compared to the injection of WSSV517 dsRNA [7]. The varied knockdown efficiency led to different protective effects, which were speculated to be associated with the functionality of the gene-encoded proteins and their roles throughout the viral life cycle [56]. Given this, shrimp feed containing only WSSV517 dsRNA might be a promising option for future applications.” 

Reviewer #2: This MS describes the use of computational methods to derive the 3D structures of WSSV051 and WSSV517 viral proteins, and identify their interacting partners. In addition, animal experiment was performed to assess the protective effects of the combined WSSV051 and WSSV517 dsRNAs against WSSV infection in P. monodon. The dsRNA incorporated feed shows stable expression of the viral genes after prolonged storage. This is an interesting study that explores the structural and functional aspects of the viral proteins and the use of RNAi could have a positive impact on shrimp disease management. The MS is well organized; however, the following suggestions would improve the manuscript.

 We appreciate your supportive comments and insightful suggestions to further improve our manuscript. We have revised the manuscript and provided a detailed response to each point, as described below. 

Specific Comments:

 1. In the material and methods (Lines 136-137) could the authors explain why the WSSV517 plasmid was re-constructed for this study whereas the WSSV051 plasmid was used from previous studies?

Response 1: Based on our previous work (Thammasorn et al., BMC Biotechnol. 2015, 15: 110), we studied the effects of dsRNA specific for WSSV VP28 and WSSV051. As a result, the WSSV051 construct had already been prepared, whereas the WSSV517 plasmid had not been previously produced. Therefore, we had to construct the WSSV517 plasmid for this study. To improve clarity, the Methods section has been edited to read as “In a previous study, a hairpin-dsRNA expression plasmid targeting one of the hubs, WSSV051, was already constructed [13]. The present study aimed to investigate the effect of combined hub dsRNA, targeting WSSV051 and WSSV517. Thus, the previously made WSSV051 construct was used, and a new hairpin-dsRNA expression plasmid targeting WSSV517 was created in this study.” (Lines 154-158).

2. For dsRNA incorporated shrimp feed preparation, what was the rationale for choosing 60 mg and 120 mg dsRNA concentrations?

Response 2: According to our previous work (Thammasorn et al., BMC Biotechnol. 2015, 15: 110), we investigated the antiviral effect of a combination of VP28 and WSSV051 dsRNA. We prepared the feed formulation using dsRNA dosages of 6 mg and 12 mg/kg of feed, which demonstrated a delay in shrimp mortality. The present study applied a ten-fold increase in dsRNA dosage in order to enhance protection against WSSV infection. This rational has now been provided in the Methods to read as “These dsRNA dosages were 10 times higher than those used in our previous study [13], which utilized 6 mg and 12 mg, with the aim of enhancing protection against WSSV infection.” (Lines 182-184).

3. Lines 170-171, the primer information for Xba1-WSSV051-RNAi-F/ Xba1-WSSV051-RNAi-R3 is not included in Table 1.

Response 3: Thank you for your indication. We have now revised the corrected primer information as indicated in Lines 191-192.

4. In Line 180, how many shrimp were used for each experimental group?

Response 4: As stated in Lines 204-207, we divided one hundred and twenty shrimp into four groups (30 shrimp per group) and used them for feeding trials, WSSV challenges, and mortality assays. We also used another twelve shrimp, three per group, for feeding trials, WSSV challenges, and sample collection for knockdown efficacy determination. This information is in Lines 214-216. 

5. In animal experiment, the number of samples (n=3) collected at a single time point seems to be inadequate for testing the efficiency of the combined WSSV hub dsRNA. Could the authors explain why they could not include more time points or use more samples for testing?

Response 5: The number of samples (n=3) conforms to the approved animal use protocol for which we obtained permission (Lines 216-218). The clear and interpretable resulting data demonstrate the adequacy of this sample size. Additionally, our research adheres to rigorous standards established by shrimp research teams commonly. For instance, similar studies assessing knockdown efficiency utilized three shrimp specimens each for genes WSSV134 (Apitanyasai et al., Fish and Shellfish Immunol. 2018, 76: 174–182) and the nonstructural prot

---

## [Editor Report · Decision Letter 1]

16 Jul 2024

Structural Modelling and Preventive Strategy Targeting of WSSV Hub Proteins to Combat Viral Infection in Shrimp Penaeus monodon

PONE-D-24-15796R1

Dear Dr. Sangsuriya,

We’re pleased to inform you that your manuscript has been judged scientifically suitable for publication and will be formally accepted for publication once it meets all outstanding technical requirements.

Kind regards,

Kenneth Söderhäll

Academic Editor

PLOS ONE
---

## [Editor Report · Acceptance letter]

19 Jul 2024

PONE-D-24-15796R1 

PLOS ONE

Dear Dr. Sangsuriya, 

I'm pleased to inform you that your manuscript has been deemed suitable for publication in PLOS ONE. Congratulations! Your manuscript is now being handed over to our production team.

Kind regards, 

on behalf of

Dr. Kenneth Söderhäll 

Academic Editor

PLOS ONE